# CRONUS: EFFICIENT LLM INFERENCE ON HETEROGENEOUS GPU CLUSTERS VIA PARTIALLY DISAGGREGATED PREFILL

## ABSTRACT

Efficient LLM inference is critical for real-world applications, especially within heterogeneous GPU clusters commonly found in organizations and on-premise datacenters as GPU architecture rapidly evolves. Current disaggregated prefill strategies, which separate the prefill and decode stages of LLM inference across different GPUs, often suffer from suboptimal performance due to imbalances between GPU capabilities and workload demands. On the other hand, extending conventional data parallelism and pipeline parallelism to heterogeneous setups incurs high inference latencies. To address these challenges, we introduce Cronus, a novel LLM inference system designed to dynamically balance workloads across heterogeneous GPUs using partially disaggregated prefill. Cronus partitions each prefill stage and executes its initial portion on the low-end GPU, while overlapping the remaining prefill and decode stages of earlier requests on the high-end GPU. Extensive evaluations across various high-end and low-end GPU combinations demonstrate that Cronus significantly improves the throughput over disaggregated prefill. It also reduces TTFT P99 and TBT P99 significantly over DP and PP while maintaining similar or better throughput.

## 1 INTRODUCTION

The proliferation of large language models (LLMs) has revolutionized various fields, enabling applications ranging from natural language processing to code generation. However, the computational demands of running these complex models, particularly during the inference phase, present significant challenges. As LLMs continue to grow in size and complexity, efficient inference becomes crucial for deploying them in real-world applications. Traditionally, LLM inference relies heavily on powerful and expensive GPU resources. With rapid innovation of GPUs (nvidia hopper architecture), newer generations of GPUs are introduced to the market in short release cycles. Yet, their high cost and limited supply have dis-incentivized cloud vendors and private organizations from retiring older generations of GPUs. As a result, these organizations are increasingly operating highly heterogeneous GPU clusters (Weng et al., 2022). Consequently, optimizing LLM inference on these heterogeneous GPU clusters has become a pressing concern.

LLM inference typically involves two primary stages: the prefill stage, where the input prompt is processed to generate the initial output token and KV cache, and the decode stage, where subsequent tokens are generated autoregressively. The prefill stage is computationally intensive, involving computations on the entire input sequence as a single large batch. In contrast, the decode stage is memory-bound (both memory bandwidth and memory capacity), as it primarily deals with retrieving and updating the KV cache for each generated token. Multiple decode requests can be batched together to improve GPU utilization, but KV cache of all requests needs to be loaded in GPU memory, requiring huge amount of memory (Yu et al., 2022a). These distinct characteristics make the prefill and decode stages suitable for different types of GPU resources. GPUs with powerful compute units are ideal for the prefill stage, while GPUs with large memory capacity and high memory bandwidth are better suited for the decode stage.

A common approach of utilizing heterogeneous GPU clusters is to employ disaggregated prefill, where the prefill and decode stages are executed on separate GPUs. However, existing disaggregated

prefill strategies (Patel et al., 2024; Zhong et al.) often struggle to achieve optimal performance due to mismatch with GPU capabilities. High-end GPUs typically offer strong computational power and large memory capacity, while low-end GPUs often offer less computational power and limited memory. If we assign the prefill stage to low-end GPUs and the decode stage to high-end GPUs, despite being more memory efficient, it is often bottlenecked by the prefill stage due to the limited computational power of low-end GPUs. Conversely, assigning the prefill stage to high-end GPUs and the decode stage to low-end GPUs can lead to the decode stage being the bottleneck due to memory limitations. In either case, existing approaches deliver low throughput due to load imbalance and resource underutilization.

Data Parallelism (DP) and Pipeline Parallelism (PP) can also be extended to support heterogeneous GPUs. DP distributes incoming requests across individual GPUs and processes them independently, while PP partitions the LLM model's layers into multiple stages, with each stage executed on a different GPU. Compared to disaggregated prefill, both approaches achieve better load balancing. However, they come with their own limitations. DP suffers from high latency when requests are routed to slower GPUs, resulting in elevated Time-to-First-Token (TTFT) and Time-between-Token (TBT). On the other hand, PP also suffers from high TTFT and lower throughput due to accumulated communication overhead between pipeline stages.

To address these challenges, we introduce Cronus, an efficient LLM inference system that dynamically balances workload across heterogeneous GPUs using partially disaggregated prefill. Cronus employs a hybrid approach to leverage the distinct capabilities of high-end and low-end GPUs. Instead of assigning each stage entirely to one type of GPU, Cronus partially executes the prefill stage on the low-end GPU, while overlapping the remaining prefill and decode stages of earlier requests on the high-end GPU. Cronus intelligently determines the optimal partition point for each prefill stage, taking into account the computational capacity of the GPUs and the characteristics of the input requests. This hybrid approach maximally utilizes both high-end and low-end GPUs and mitigates the load imbalance issue encountered in existing disaggregated prefill strategies. Furthermore, by partially executing the prefill stage on the low-end GPU, Cronus reduces the TTFT compared to fully assigning the prefill stage to the low-end GPU (DP) and without accumulated prefill overhead (PP). We conduct extensive evaluations across multiple heterogeneous GPU combinations, demonstrating that Cronus significantly improves the throughput over disaggregated prefill. It also reduces TTFT P99 and TBT P99 significantly over DP and PP while maintaining similar or better throughput.

## 2 BACKGROUND

**LLM Inference** The inference of most popular LLM models, *e.g.,* the GPT (Brown et al., 2020) and LLaMA (Touvron et al., 2023a) series, is done in an autoregressive manner, which consists of two stages: the *prefill* stage, where the user prompt is processed to generate the first token of the response, and the *decode* stage, where subsequent tokens are generated one by one until a special end-of-sequence token is reached. Both stages run the same LLM model, which consists of multiple (32 for LLaMA-2 7B (Touvron et al., 2023b)) Transformer blocks, and each Transformer block is in turn composed of an attention component and an MLP component (Vaswani et al., 2017). The LLM model only runs once during the prefill stage, where tokens from the user prompt are processed in a batch, which is very compute-intensive. On the other hand, during the decode stage, the model runs once for each output token. However, with the widely-used KV-cache optimization, only the last token needs to be processed by the model in order to generate the next token. Thus, the model essentially runs with batch size 1 and is memory-intensive instead of compute-intensive.

Various approaches have been developed to optimize the system by leveraging the different characteristics of two stages. Continuous batching (Yu et al., 2022b) construct batches with decodes from different request in-flight. It allows the decode of new requests batches with decode of old requests, increasing the batch size of the decode iteration, improving inferencing efficiency. Chunked prefill (Agrawal et al., 2024; 2023) splits prefill of a request into multiple chunks, and batches compute-intensive chunked prefill with memory-intensive decode. Disaggregated prefill (Zhong et al.) processes the prefill and the decode of a request on different engines with different configuration and hardware.

The QoE of the two stages are also measured separately. The latency of the prefill stage is measured as the time-to-first-token (TTFT), while the latency of generating one token in the decode stage is

Table 1: Comparison of previous approaches.

| Approach | Load Balance | Communication | Batch Size | TTFT P99 | TBT P99 | Throughput |
|---|---|---|---|---|---|---|
| Disagg. H-L | Poor | KV cache | Small | Small | Small | Low |
| Disagg. L-H | Poor | KV cache | Large | Large | Large | Low |
| DP+Chunked | Good | No | Large | Large | Large | High |
| PP+Chunked | Good | Every iteration | Small | Large | Large | Low |
| Cronus | Good | Partial KV cache | Large | Medium | Medium | High |

measured as the time-between-tokens (TBT). 99th percentile TTFT (TTFT P99) and 99th percentile TBT (TBT P99) are two common metrics used to evaluate the performance of the inferencing engine (Patel et al., 2024; Agrawal et al., 2024). They capture the latency in the worst 1% scenario.

## 3 PREVIOUS APPROACHES AND LIMITATIONS

We discuss the advantages and limitations of existing approaches to LLM inference serving on heterogeneous GPU clusters, as show in Table 1.

### 3.1 DISAGGREGATED PREFILL

The disaggregated prefill scheme (Patel et al., 2024; Zhong et al.) was motivated by the observation that the prefill and decode stages exhibit different performance characteristics, and was proposed to decouple the optimization of these two stages. Specifically, because the prefill stage is compute-bound, it should be executed on GPUs with high computational capacity. In contrast, the decode stage is memory-bound and is better suited for GPUs with large memory and high memory bandwidth.

In practice, most high-end GPUs provide not only high computational capacity but also larger memory and higher memory bandwidth, while most low-end GPUs have lower compute power, smaller memory, and lower memory bandwidth. This heterogeneity poses a challenge for deploying disaggregated prefill: either the prefill stage is assigned to a GPU with limited memory but also insufficient compute capacity, or the decode stage is assigned to a GPU with lower compute power but also limited memory. Neither configurations are optimal. We next explain in detail the implications of both configurations.

**Disaggregated Low-High.** If the prefill stage is processed by the low-end GPU and the decode stage is processed by the high-end GPU, the inference will suffer from large TTFT due to the low compute capacity of the low-end GPU. Moreover, for requests with long input lengths and short output lengths, the prefill stage achieves lower throughput than the decode stage, making it the bottleneck of the pipeline.

**Disaggregated High-Low.** Since the decode stage runs on the low-end GPU, the available memory for the KV cache is limited. Due to this memory constraint, for certain workloads, even fully utilizing the low-end GPU's memory may not provide sufficient throughput to match the prefill stage. On the other hand, as the prefill stage is processed by high-end GPUs, the prefill stage achieves higher throughput than the decode stage, and the prefill GPU becomes periodically idle while waiting for the decode stage to process requests. This leads to increased latency, reduced GPU utilization, and lower maximum throughput.

### 3.2 DATA PARALLELISM + CHUNKED PREFILL

One straightforward approach to leveraging heterogeneous GPUs is combining data parallelism with chunked prefill: individual GPUs process reqeusts independently, while a front-end dispatcher distributes incoming requests across them (Chunked prefill is used to avoid spikes in TBT when new requests are processed). With no inter-engine communication, this approach incurs minimal overhead. However, data parallelism has a clear drawback: requests routed to a slower GPU will experience higher latency. Consequently, the GPU cluster exhibits high TTFT P99 and TBT P99.

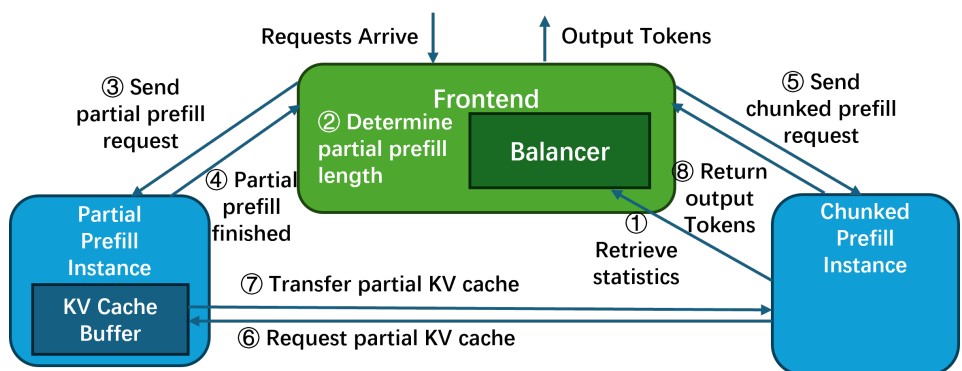

Figure 1: Overview of partially disaggregated prefill.

### 3.3 PIPELINE PARALLELISM + CHUNKED PREFILL

An alternative approach to leveraging heterogeneous GPUs is combining pipeline parallelism with chunked prefill, where a single inference engine partitions the model's layers into multiple stages, with each stage executed on a different GPU in a pipelined fashion. The number of layers assigned to each stage is tuned to match the computational capacity of the GPU allocated to that stage.

Compared to data parallelism, pipeline parallelism introduces several overheads. First, it incurs additional communication overhead between pipeline stages. While the communication overhead is incurred once for each generated token in the decode phase, as a prefill request is divided into chunks, the communication overhead is incurred once for each chunk, which adds up and significantly increases the TTFT.

A second overhead of a heterogeneous GPU pipeline is the reduced batch size for decode requests compared to using only the high-end GPU. Pipeline parallelism splits requests into N batches where N equals to the number of pipeline stages. When a high-end large-memory GPU forms a pipeline with a low-end small-memory GPU, through total available GPU memory being larger than a single high-end GPU, the effective memory size for each batch becomes smaller, resulting in smaller batch sizes. The reduced batch size in turn reduces the computational efficiency of the batched inference, and lowers the throughput of each decode iteration.

## 4 CRONUS

### 4.1 KEY IDEA

As discussed in Section 3, due to the mismatch of computation and memory requirements of both LLM inference stages with the GPU design, both disaggregated prefill designs suffer from load imbalance and thus low throughput. To avoid the load imbalance, we propose to 1) run the decode phase on the high-end GPU to take advantage of its memory capacity, and 2) run the prefill phase partially on the low-end GPU while offloading the rest of prefill computation to the high-end GPU (in the form of chunked prefill) to take advantage of the extra computational capacity. Such a hybrid design is able to utilize the capacity of both GPU types and significantly boosts the inference throughput over prior disaggregated prefill designs (Table 1). Furthermore, compared to DP+chunked and PP+chunked, since our decode phase runs completely on the high-end GPU, we are able to achieve better TBT P99. We also achieve better TTFT P99 since we run prefill partially on the high-end GPU (compared to DP+chunked which runs some prefill requests completely on the low-end GPU), and we don't have the accumulated prefill overhead as in PP+chunked.

### 4.2 CRONUS OVERVIEW

Figure 1 shows the architecture of our design, Cronus. Similar to disaggregated prefill, Cronus runs a single instance of LLM inference across a pair of high-end and low-end GPUs. It includes a frontend,

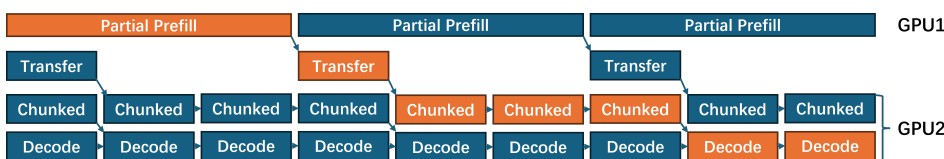

Figure 2: A request's prefill and KV cache transfer is overlapped with other requests' computation

a partial prefill instance (PPI) on the low-end GPU, and a chunked prefill instance (CPI) on the high-end GPU.

The Balancer, located in the frontend, determines how to split the prefill workload of each incoming prompt between the PPI and CPI. The KV caches generated by the PPI are stored in a KV cache buffer, where they await retrieval by the CPI for further processing.

When a new request arrives at the frontend, it waits until the waiting queue in the partial prefill instance becomes empty. At that point, the Balancer ① retrieves statistics from the chunked prefill instance, ② determines the partial prefill length—that is, the portion of the prompt to be processed by the PPI—and ③ dispatches the request to the PPI. By limiting the total number of requests in the PPI to at most two at a time, we ensure that the partial prefill length for each request is calculated using up-to-date statistics from the chunked prefill instance.

Once the PPI completes the partial prefill for a request, it stores the computed KV cache for the processed prompt segment in the KV cache buffer and ④ sends a notification to the frontend indicating that the partial prefill is complete. The frontend then ⑤ sends a chunked prefill request to the chunked prefill instance. This chunked prefill request contains the original request along with an additional field specifying the length of the prompt already processed by the PPI.

When a new request is scheduled in the chunked prefill instance, the engine first checks whether it needs to retrieve a KV cache from the partial prefill instance. If retrieval is required, the chunked prefill instance ⑥ sends the request's prompt to the partial prefill instance, which then ⑦ transfers the corresponding KV cache from the KV cache buffer to the chunked prefill instance. This KV cache transfer occurs during the first iteration of the request, replacing original computation, and overlaps with the computation of other requests' decode and/or chunked prefill stages as shown in Fig. 2. After the first iteration, the request proceeds using the standard chunked prefill process.

### 4.3 THE BALANCER

Recall that the goal of Cronus is to balance the workload between the two GPUs and achieve full utilization of both, by splitting the prefill stage of each request $R_i$ and pipeline its execution: the first part of prefill of $R_i$ runs on the first GPU, followed by the second part of prefill of $R_i$ on the second GPU. The first part of prefill overlaps with the decoding of earlier requests $(R_{i-j-1}, ..., R_{i-j-k})$, as shown in Figure 2.

A critical challenge in this design is determining the optimal prefill split for each incoming request $R_i$ to maintain balanced load and ensure both GPUs remain fully utilized.

We use a simple heuristic to balance the load across the two GPUs in the pipeline: optimal balance is achieved when all pipeline stages have the same throughput. We denote the two parts of prefill of each request $R_i$ as $R_i^{P1}$ and $R_i^{P2}$, and its decode as $R_i^D$; $R_i^{P1}$ will run on GPU 1 (partial prefill instance) and and $R_i^{P2}$ and $R_i^D$ will run on GPU 2 (chunked prefill instance). And the throughput of two stages is guaranteed to be the same when the execution time $T_{parprefill}$ of partial prefill $R_i^{P1}$ on GPU 1 equals the execution time $T_{chunked}$ of chunked prefill finishing $R_i^{P2}$ on GPU 2.

To estimate these execution times, we build two predictors. The first predictor estimates $T_{parprefill}$, and the second predictor estimates the execution time $t_{chunked}$ of a single iteration in the chunked prefill instance on GPU 2, which batches $R_i^{P2}$ and decoding of previous requests.

To calculate the total execution time $T_{chunked}$ of the chunked prefill on GPU 2, we need to estimate the execution time of each chunked prefill iteration, $t_{chunked}$, and calculate the sum of them. In Section 4.4, we model $t_{chunked}$ as a linear function of prefill context length. Then $T_{chunked}$ becomes

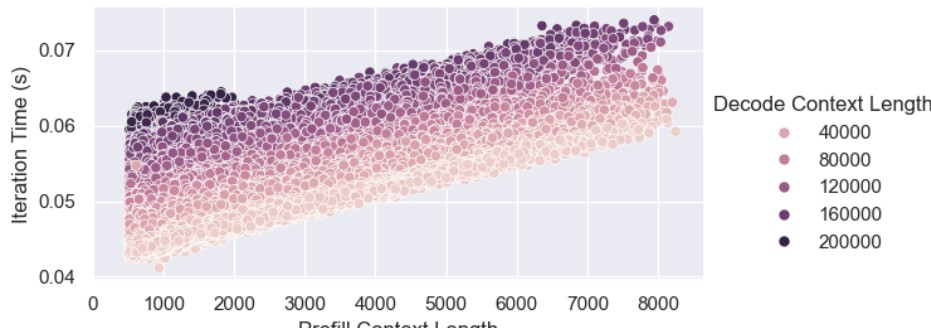

Figure 3: Time of one chunked prefill iteration on A100 with LLaMA3-8B.

the sum of an arithmetic sequence using equation 1.

$$T_{chunked} = N_{iter} \frac{t_{chunked(first)} + t_{chunked(last)}}{2} \tag{1}$$

where $N_{iter}$ is number of chunked prefill iteration required to finished part 2 of the request $R_i$, $t_{chunked(first)}$ and $t_{chunked(last)}$ are execution times of the first and the last chunked prefill iteration of $R_i$.

## 4.4 MODELING PARTIAL PREFILL AND CHUNKED PREFILL EXECUTION TIME

**Partial prefill**  Since the partial prefill instance runs prefill for one request at a time, the execution time of partial prefill solely depends on the chosen partial prefill length. Therefore, we model the partial prefill time $T_{parprefill}$ as a function of the partial prefill length using Equation 2,

$$T_{prefill}(R_i^{P1}) = k_p \cdot L(R_I^{P1}) + b_p \tag{2}$$

where $L(R_i^{P1})$ is the length of $R_i^{P1}$, and $k_p$ and $b_p$ are coefficients from linear regression on profiled data. The Linear fit of prefill execution time of LLaMA3-8B model on A30 achieves R2 score of 0.993 and mean absolute percentage error of 7.4%.

**Chunked prefill**  Modeling the execution time of a single chunked prefill iteration $t_{chunked}$ is more complex, as it involves a batch containing both prefill and decode requests.

The majority of the execution time comes from the MLP layers and attention layers. The MLP execution time depends only on the number of batched input tokens. Since the chunk size is configured as a constant in our system, The input size of the MLP layer is fixed. So, the MLP execution time can be treated as a constant.

The attention layer execution time consists of two components: attention for decode requests and attention for chunked prefill. Since at the time we split request $R_i$ we cannot foresee whether previous requests $(R_{i-1}, ..., R_{i-j-k})$ have finished or not when $R_i^{P2}$ starts, we assume the system is stable and the same decode requests stay there when $R_i^{P2}$ starts.

Decode attention involves matrix-vector multiplications, which are highly memory-bound and can be modeled as a function of the total size of the decode requests. Prefill attention, in contrast, involves matrix-matrix multiplications. Its execution time depends on the prefill context length and the number of prefill tokens, since these parameters determine the matrix sizes in the attention operation. Therefore, prefill attention execution time can be modeled as a function of these two factors.

We model the chunked prefill iteration time $t_{chunked}$ using Equation 3,

$$t_{chunked} = k_{ctxp} \cdot L(R_i^{P2}) + k_{ctxd} \cdot \sum_l L(R_l^D) + b_c \tag{3}$$

where $L(R_i^{P2})$ is the context length of the second part of prefill request $R_i$ in this chunked prefill iteration, and $L(R_l^D)$ is the context length of decode request $R_l^D$ in the batch, and others are

coefficients from linear regression on profiled data. The number of prefill tokens is not considered in the equation, because even though it varies between iterations, the variation is insignificant as they are always approximately equal to the maximum number of batched tokens.

Figure 3 presents the measured chunked prefill iteration time for LLaMA3-8B on an A100 GPU. Each iteration uses 512 batched tokens. The y-axis shows the iteration time, the x-axis represents the prefill context length, and the hue of the data points indicates the decode context length. The linear fit achieves an $R^2$ score of 0.990 and a mean absolute percentage error of 0.8%.

## 4.5 IMPLEMENTATION

We implemented Cronus on top of a developing branch of vLLM (Kwon et al., 2023) version 0.6.1.post2 (Apache License 2.0). The details are included in the supplemental material.

## 5 EVALUATION

In this section, we compare throughput, TTFT P99, TBT P99 of Cronus and 4 baselines using two models and two hardware configurations.

### 5.1 EVALUATION SETUP

**Hardware Environment:** We evaluate our design using two different heterogeneous GPU combinations — A100 (80 GB) + A10 (24 GB) and A100 (80 GB) + A30 (24 GB). For both setups, the GPUs are on different nodes and are connected using InfiniBand (100Gbps). For each node, we use 4 CPU cores and 40 GB of memory.

**Datasets:** We use conversation traces from Microsoft's Azure LLM inference trace 2023 used in Microsoft's splitwise paper (Patel et al., 2024). These traces are licensed under the CC-BY Attribution License. To reduce the overall time of benchmarking, we use 1000 traces in each of our experiments. Requests are sent to the inferencing engine or the frontend with fixed time interval. The average input length and output length of the conversation traces we used are 1014 and 247.

**Models:** We evaluate our design with LLaMA3-8B (Grattafiori et al., 2024) (license: META LLAMA 3 COMMUNITY LICENSE AGREEMENT) and Qwen2-7B (Yang et al., 2024) (license: Qwen LICENSE AGREEMENT). For pipeline parallelism, to balance the load among heterogeneous GPUs, model layers are unevenly split between two GPUs based on their BFloat16 FLOPS. LLaMA3-8B has 32 layers. It is split into 23 and 9 layers on A100+A10 cluster, and into 21 and 11 layers on A100+A30 cluster. Qwen2-7B has 28 layers. It is split into 20 and 8 layers on A100+A10 cluster, and into 18 and 10 layers on A100+A30 cluster.

**Metrics:** We evaluate the performance of our design over the baslines in three dimensions: throughput, TTFT P99, and TBT P99.

**Baselines:** We compare Cronus with 4 baselines: pipeline parallelism in vLLM version 6.1, data parallelism with a weights round-robin, Disaggregated prefill High-Low, and Disaggregated prefill Low-High. In data parallelism we assign a weight of 3 to A100 and a weight of 1 to A10 or A30 and we limit the number request in the waiting queue of A100 to 3 and of A10 or A30 to 1. For disaggregated prefill, we use the same code as our partial prefill implementation, but always set the partial prefill length to the input length. To have a fair comparison between disaggregated prefill and other approaches, their TTFT includes the KV cache transfer time. We enable chunked-prefill in pipeline parallelism and data parallelism. For all baselines utilizing chunked prefill, we set the maximum token batch size to be 512, except for DP requests running on A10 or A30, where we use a smaller chunked size 256 to reduce the difference of TBT on low-end and high-end GPUs. We set the maximum token batch size of chunked prefill instance in Cronus to 512.

### 5.2 THROUGHPUT

Table 2 shows the maximum throughput of Cronus and other 4 baselines. We measure the maximum throughput by sending all request at the start and then measuring the throughput of the system. Cronus has significantly high throughput than PP (up to 2.58×), Disaggregated High-Low (up to

Table 2: Maximum throughput (request per second)

| Approach | A100+A10 LLaMA3-8B | A100+A10 Qwen2-7B | A100+A30 LLaMA3-8B | A100+A30 Qwen2-7B |
|---|---|---|---|---|
| DP+Chunked | 7.28 | 8.70 | 8.54 | 10.85 |
| PP+Chunked | 3.86 | 4.08 | 3.96 | 3.97 |
| Disagg. H-L | 1.31 | 3.45 | 2.93 | 6.74 |
| Disagg. L-H | 4.11 | 4.35 | 6.14 | 6.59 |
| Cronus | 7.39 | 8.29 | 8.7 | 10.27 |

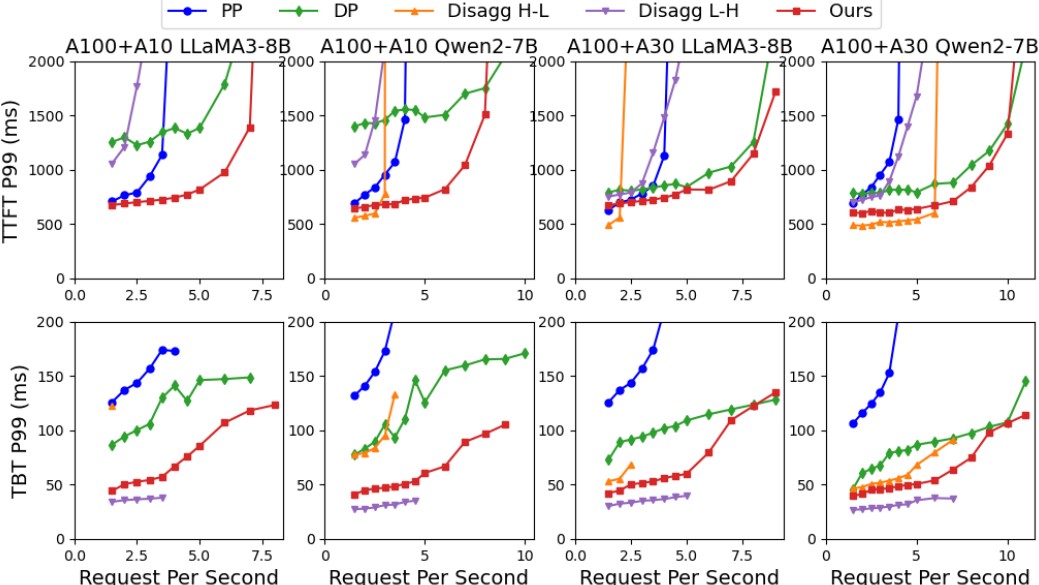

Figure 4: TTFT P99 and TBT P99 of conversation traces using different hardware and models

5.64×), and Disaggregated Low-High (up to 1.9×). The communication overhead of PP and the small batch size contribute to the low throughput of PP. The imbalance between prefill and decode instance in disaggregated High-Low and Disaggregated Low-High contributes to low throughput of these disaggregated approaches. We demonstrate the load imbalance of disaggregated prefill through an experiment in Appendix B. The maximum throughput of Cronus and the maximum throughput of DP are comparable as both achieves good utilization of heterogeneous GPU resources.

### 5.3 TTFT P99

The first row of Fig. 4 compares the TTFT P99 between Cronus and 4 baselines on different hardware and different models.

Cronus achieves up to 55% lower TTFT P99 than DP on A100+A10 hardware and up to 26% lower TTFT P99 than DP on A100+A30 hardware. TTFT P99 of DP increases significantly when A30 is downgraded to A10 as TTFT P99 is more sensitive to low-end GPU's performance. In contrast, all requests in Cronus can benefit from the compute power of both GPUs, and their TTFT P99 is less sensitive to low-end GPU's performance.

Cronus achieves up to 84% lower TTFT P99 than Disaggregated Low-High. Disaggregated Low-High processes all prefill on low-end GPU which introduces large TTFT P99. Cronus avoids this problem by distributing the workload of prefill stage between the low-end GPU and the high-end GPU, resulting in better TTFT P99 and throughput.

Cronus achieves up to 58% lower TTFT P99 than PP. Although both Cronus and PP utilized both GPUs processing prefill, PP has more overhead in each iteration and requires more iterations to complete the prefill. For requests with a long input, overhead in PP accumulates and increases above the KV cache transfer time in Cronus by a significant margin.

Disaggregated High-Low is the only baseline whose TTFT P99 is consistently lower than TTFT P99 of Cronus. In fact, Disaggregated Low-High attains the best TTFT P99 possible since the high-end GPU is dedicated to prefill only. However, the TTFT advantage comes with a heavy cost: significantly lower throughput compared to load-balanced approaches like DP and Cronus, making it impractical to use in real-world scenarios.

### 5.4 TBT P99

The second row of Fig. 4 compares the TBT P99 between Cronus and 4 baselines on different hardware and different models. Cronus obtains up to 70% smaller TBT P99 than PP, up to 63% lower TBT P99 than DP, and up to 51% than Disaggregated High-Low. TBT P99 of PP suffers from communication overhead, while DP and Disaggregated High-Low process the decode of some or all the requests in low-end GPU which slows down the decode stage and increases TBT P99 significantly (even if decode requests in Disaggregated High-Low are not piggybacked with prefill requests). Cronus, on the other hand, processes all requests' decode on high-end GPU. Disaggregated Low-High has the best TBT P99 because it dedicates the high-end GPU to only decode. However, similar to Disaggregated High-Low for TTFT P99, this approach is severely imbalanced and is impractical for real-world scenarios.

## 6 LIMITATIONS

Even though Cronus processes decode on high-end GPU, the high-end GPU can still be bottlenecked by the decode phase when all the requests have short input lengths and long output lengths. In such a case, high-end GPU may have lower throughput than the low-end GPU even if all the prefill is processed in the low-end GPU and Cronus may experience load imbalance. The load imbalance can be mitigated by offloading some decode requests to the prefill node, which we plan to explore as future work.

## 7 RELATED WORK

Several recent work studies supporting LLM inference on heterogeneity GPU clusters via partitioning and scheduling. HexGen(Jiang et al., 2024) employs asymmetric parallelism, assigning larger model segments to faster GPUs, while slower or memory-rich GPUs handle lighter or memory-intensive workloads. However, in practice, faster GPUs often also possess greater memory capacity. LLM-PQ (Zhao et al., 2024) introduces phase-aware partitioning and adaptive quantization, aligning precision and partition size with each GPU's capabilities. Cost-aware approaches like Mélange (Griggs et al., 2024) leverage heterogeneity to reduce inference costs by dynamically assigning requests based on each GPU's price-performance characteristics. These works are orthogonal to ours, which directly tackles the load imbalance of disaggregated prefill on heterogeneous GPUs.

## 8 CONCLUSION

We presented Cronus, an efficient LLM inference system that dynamically balances workload across heterogeneous GPUs using partially disaggregated prefill. We conducted extensive evaluations across multiple heterogeneous GPU combinations, demonstrating that Cronus significantly improves the throughput over disaggregated prefill by up to 5.64×. In addition, it reduces TTFT P99 by up to 55% over DP and up to 58% over PP, reduces TBT P99 by up to 63% over DP and up to 70% PP while maintaining similar or better throughput.

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

## A   BALANCER ALGORITHM DETAIL

---

**Algorithm 1** Balancer algorithm

---

**Input:** $L_{in}$ prompt length,
**Input:** $k_p$ and $b_p$: prefill execution time model parameter
**Input:** $k_{ctxp}$, $k_{ctxd}$, and $b_c$: chunked prefill execution time model parameter
$\quad n_p \leftarrow$ number of requests in chunked prefill instance
$\quad L_{ctxp} \leftarrow$ sum of all requests' context length in chunked prefill instance
$\quad N_{size} \leftarrow$ KV block size in chunked prefill instance
$\quad B \leftarrow$ Maximum number of batched tokens in each iteration
$\quad$**if** $N_{free} < \left\lceil \frac{L_{in}}{N_{size}} \right\rceil$ **then**
$\quad\quad$use $L_{in}$ as the partial prefill length
$\quad$**else**
$\quad\quad L_p \leftarrow (\left\lceil \frac{1}{512}L_{in} \right\rceil, \left\lceil \frac{2}{512}L_{in} \right\rceil, \ldots, \left\lceil \frac{512}{512}L_{in} \right\rceil)$
$\quad\quad T_{prefill} \leftarrow k_p L_p + b_p$ $\qquad\qquad\qquad$ ▷ Estimate partial prefill time
$\quad\quad n_p \leftarrow B - n_d$
$\quad\quad L_c \leftarrow L_{in} - L_p$
$\quad\quad N_{iter} \leftarrow \left\lceil \frac{L_c}{n_p} \right\rceil$ $\qquad\qquad\qquad$ ▷ Calculate the number of chunked prefill iteration
$\quad\quad L_{last} \leftarrow L_p + \left\lfloor \frac{L_c}{n_p} \right\rfloor n_p$ $\quad$ ▷ Calculate the prefill context of the last chunked prefill iteration
$\quad\quad T_{chunked} \leftarrow N_{iter} \left( k_{ctxp} \frac{L_{in} + L_{last}}{2} + k_{ctxd} L_{ctxd} + b_c \right)$ ▷ Estimate total chunked prefill time
$\quad\quad idx \leftarrow \text{argmin} \left( |T_{prefill} - T_{chunked}| \right)$
$\quad\quad$use $L_p[idx]$ as the partial prefill length
$\quad$**end if**

---

Algorithm 1 is the algorithm implemented in the balancer. It first checkes whether CPI have enough free KV blocks to receive the requests. If CPI does not have enough free KV blocks, the request's input will be process only in PPI, so the partial prefill length of the request is set to the input length. If CPI has enough free KV blocks, algorithm generates some candidate partial prefill length by evenly sample between 1 and input length. Then estimate the partial prefill time and the total chunked prefill time of the request for each partial prefill length candidate. Use the candidate with smallest absolute difference in partial prefill time and total chunked prefill time as the partial prefill length of the request.

## B   LOAD IMBALANCE IN DISAGGREGATED PREFILL

To demonstrate the load imbalance in Disaggregated High-Low and Disaggregated Low-High, we measure relative GPU utilization rate shown in Table 3. The relative GPU utilization rate is calculate

Table 3: relative GPU utilization rate in disaggregated prefill

| Approach | Disagg. H-L | | Disagg. L-H | |
| --- | --- | --- | --- | --- |
| Configuration | Prefill | Decode | Prefill | Decode |
| A100+A10 LLaMA3-8B | 11% | 97% | 99% | 32% |
| A100+A10 Qwen2-7B | 28% | 101% | 104% | 25% |
| A100+A30 LLaMA3-8B | 25% | 96% | 98% | 47% |
| A100+A30 Qwen2-7B | 54% | 100% | 99% | 38% |

by dividing the maximum throughput of the whole system by the maximum throughput of the decode or prefill instance. For example, to measure the prefill GPU utilization, we divide the overall throughput by the maximum prefill throughput of the prefill instance.

As shown in Table 3 the low-end GPU in both Disaggregated High-Low and Disaggregated Low-High has a utilization rate around 100% and the high-end GPU only has at most 54% utilization rate. The imbalance is more severe when the low-end GPU is weaker.

