# OpenReview forum: "Cronus: Efficient LLM inference on Heterogeneous GPU Clusters via Partially Disaggregated Prefill"
_ICLR.cc/2026/Conference — Submitted to ICLR 2026_

### Official Review · Reviewer_URKn · 2025-10-31

**Soundness:** 3
**Presentation:** 2
**Contribution:** 2
**Rating:** 0
**Confidence:** 5

**Summary:**

This paper presents Cronus, a system for efficient LLM inference on heterogeneous GPU clusters, with a particular focus on configurations involving one high-end and one low-end GPU. The key proposal is "partially disaggregated prefill", in which the prefill stage of inference is split between the two GPUs: part of the prefill is executed on the low-end GPU, while the remainder (along with the decode phase) is handled by the high-end GPU. In addition, the authors provide a simple performance model for estimating execution time on both GPU types. Experiments compare Cronus against data parallelism, pipeline parallelism, and existing disaggregated prefill strategies using two LLMs (LLaMA3-8B and Qwen2-7B) and two GPU pairings. The results show that Cronus improves throughput as well as the 99th-percentile Time-To-First-Token (TTFT) and Time-Between-Tokens (TBT) compared to existing methods.

**Strengths:**

- The paper addresses a common problem in LLM deployment, where heterogeneous GPU clusters are utilized. The inefficiencies of conventional prefill/decode separation and parallelism methods in such settings are convincingly motivated.

- The proposed partially disaggregated prefill approach is well-justified. Unlike prior works that fully assign prefill or decode to specific devices, Cronus adaptively splits and schedules prefill operations to maximize GPU utilization while balancing compute and memory constraints.

- The system mechanics are described with adequate precision. The paper provides a load-balancing heuristic (Section 4.3), explicit timing models (Equations 2 and 3), and an ablation study explaining how prefill and chunked prefill times are profiled and predicted.

- The evaluation is thorough, including two real LLMs of different sizes, two heterogeneous cluster setups (A100+A10 and A100+A30), and real workload traces from Azure LLM inference logs. The results show that Cronus achieves the best of both worlds, consistently demonstrating superior throughput, TTFT P99, and TBT P99 performance curves.

**Weaknesses:**

- While the paper introduces linear models for prefill and chunked prefill prediction, these models are derived from limited profiling and lack deeper statistical motivation or robustness analysis. Equation (3) accounts for context length, but the effects of real-world queueing, arrival rate variance, or workload burstiness are not theoretically addressed.

- The experiments rely on a single real-world trace with average input and output lengths. There is insufficient exploration of other workloads, particularly those with varying input/output ratios, highly bursty traffic, or adversarial scenarios (e.g., predominantly long outputs or very short prompts).

- Cronus is validated only on a pair of high-end and low-end GPUs. The paper does not discuss scalability to larger clusters, arbitrary multi-GPU topologies, or configurations involving more than two device classes. Although the system is described as “dynamically balancing workloads,” it remains unclear how this approach generalizes to clusters with more than two heterogeneous GPUs.

- The front-end (with the Balancer) must orchestrate queueing, dispatching, prediction, and transfer notifications for each request, as illustrated in Figure 1. The computational and synchronization overheads of the Balancer are not quantified. In real-world high-throughput settings, front-end coordination, especially notification handling, batch assembly, and queue state refreshing, could introduce non-negligible latency.

- Minor errors: (1) Line 107 mentions "QoE" without explanation; (2) Line 158 contains a typo ("reqeusts"); (3) Line 320 lacks a definition for $R_l^D$.

- Although the reviewer appreciates the useful strategy proposed for handling heterogeneous GPU settings, the paper’s format is incorrect (it lacks the statement “Under review as a conference paper at ICLR 2026” in the header). As a result, the reviewer must assign a score of 0 for formatting compliance.

**Questions:**

- How would Cronus generalize to clusters with more than two GPUs or more than two types/classes of GPUs? Is the current design readily extensible to K heterogeneous devices, or are there architectural bottlenecks that limit scalability?

- How robust is the Balancer to errors in its predictive models? What is the performance impact if queue statistics, chunk timing, or device throughput are inaccurately estimated?

- Can the authors provide empirical or analytical evidence on the latency and computational overhead incurred by the front-end Balancer? Is this overhead negligible under high-load conditions, or could it become a bottleneck in practice?

- Under high batch concurrency or network contention, have any bottlenecks been observed in partial KV-cache transfers? How does Cronus handle degraded network performance in such cases?

- Have the authors evaluated Cronus under more diverse or adversarial workloads, for example, scenarios with extreme input/output length ratios, non-stationary request arrivals, or multi-tenant LLM inference (serving multiple models or user groups in parallel)?

---

### Official Review · Reviewer_rtB8 · 2025-11-01

**Soundness:** 2
**Presentation:** 3
**Contribution:** 2
**Rating:** 2
**Confidence:** 4

**Summary:**

The paper proposes Cronus, a serving system for heterogeneous GPU clusters that partially disaggregates prefill: a short prefix of prefill runs on a low-end GPU, then the remainder of prefill plus all decoding run on a high-end GPU, overlapping compute and KV transfers. A predictor chooses the split per request. Experiments on A100+A10 and A100+A30 with LLaMA3-8B and Qwen2-7B report higher throughput than disaggregated prefill and lower TTFT/TBT P99 than DP and PP.

**Strengths:**

1. This paper tries to tackle an important problem of how to improve LLM serving efficiency on heterogeneous clusters.

2. There is a good articulation of the key design of the system that is based on partial disaggregation of prefill and decoding.

3. The paper reports max throughput close to existing DP baselines and gains on P99 TTFT and P99 TBT compared with baselines.

**Weaknesses:**

1. The paper misses a thorough technical discussion and evaluation with the baselines. Moreover, the motivation part is weak as only conceptual comparisons are given.

2. It is unclear why the system chooses to support DP and PP but not TP, given that TP is so widely used in LLM serving workloads these days.

3. There is no evaluation showing that the approach could scale up with a larger number of nodes in the cluster or a pool of GPUs.

4. Evaluation is mainly based on a single dataset.

**Questions:**

It is not clear to me both conceptually and technically why Cronus is a better approach compared to existing state-of-the-art LLM serving disaggregation methods such as Splitwise and DistServe.
The paper mentions that existing systems often struggle to achieve optimal performance due to a mismatch with GPU capabilities.
However, existing systems are neither evaluated in the previous approach section nor in the evaluation section.
The previous approach section is mainly explaining conceptual weaknesses of simpler baselines such as disaggregating prefill to high-end GPUs and decode to low-end GPUs and vise versa.
I think the baselines are an oversimplification of existing methods.
For example, depending on the workload, Splitwise has a mixed pool of GPU instances that runs mixed batches of prefill and decode.
Also, DistServe has optimizations on the batching strategy such that it controls the batch size of prefill instances to prevent making prefill more compute-bound.
Without actual comparisons with such methods, it is hard to evaluate the benefits of the paper.

Also, the system seems to not support TP, which is widely used in LLM serving.
TP is shown to have better throughput performance than using PP under the same setup, since PP may introduce extra latency due to communications between stages.
Therefore, it is a bit confusing why the system only supports and targets the optimizations on DP and PP alone but neglect TP.

The evaluation setup is limited. For the hardware setup, the design analysis and evaluation are limited to only two GPUs.
It is not clear how the system could scale as the number of GPUs increases.
Also, only one dataset Splitwise is evaluated.
It would be beneficial to evaluate the system performance on other datasets that have different request characteristics.


1. Can we show evaluation results comparing with existing state-of-the-art methods such as Splitwise and DistServe?

2. How can the current system handle burstiness in the workload?

3. How would the system support TP and what will be the throughput and latency if TP is used?

4. In Section 4.2, "By limiting the total number of requests in the PPI to at most two at a time, ...", what does that mean? Why is the batch size of PPI limited to 2?

5. How does the system scale with a different number of GPUs or nodes? How would the system perform under other datasets with different input and output distributions that lead to different compute and memory characteristics? Can the system handle long-context workloads?

---

> ### Author Response · Authors · 2025-11-29
>
> **Q: Can we show evaluation results comparing with existing state-of-the-art methods such as Splitwise and DistServe?**
>
> **A:** Due to limited accessibility of hardware, we could not find large heterogenous clusters with A100 and A10/A30. So, we were not able to run experiments on a large heterogeneous cluster.
>
> **Q: How can the current system handle burstiness in the workload?**
>
> **A:** The design of Cronus focuses on maximizing the throughput of a single inference instance. We expect the burstiness of traffic served will be handled by an autoscaler on top of Cronus at the cluster level.
>
> **Q: How would the system support TP and what will be the throughput and latency if TP is used?**
>
> **A:** Our frame naturally extends to TPs, e.g., TP of weak GPUs to perform partial prefill, and TP of strong GPUs to perform the remaining prefill batches with decode. We will add such experiments in the final paper.
>
> **Q: In Section 4.2, "By limiting the total number of requests in the PPI to at most two at a time, ...", what does that mean? Why is the batch size of PPI limited to 2?**
>
> **A:** We limit the waiting queue of partial prefill instances to 2 so that each prefill can have a split ratio based on more up-to-date information from the balancer.
>
> **Q: How does the system scale with a different number of GPUs or nodes?**
>
> **A:** Cronus is designed to scale to X high-end GPUs and Y low-end GPUs. Because of hardware accessibility limitations, we could not conduct several experiments we intended in evaluation before submission. We have done experiments of more configurations and will add these results into the camera ready paper.
>
> In particular, we have run experiments on different numbers of high-end GPUs and low-end GPUs: with 2 A10+1 A100, and with 1 A10+2 A100. We used data parallelism in partial prefill instances and chunked prefill instances to allow more than one GPUs in these instances. To balance the throughput of partial prefill instances and chunked prefill instances, we simply adjusted the Balancer so that the ratio of partial prefill time over the chunked prefill time of a request matches the ratio of the number of chunked prefill instances and the number of partial prefill instances. In both configurations, Cronus showed the same advantage over other configurations mentioned in the paper.
>
> In more detail, in experiments with 2 A10+1 A100, Cronus reduces the TTFT P99 by up to 43% and TBT P99 by up to 40% compared to DP, and increases throughput by 750% compared to disagg LH and by 167% compared to disagg HL;  In experiments with 1 A10+2 A100, Cronus reduces the TTFT P99 by up to 33% and TBT P99 by up to 53% compared to DP, and increases throughput by 264% compared to disagg LH and by 41% compared to disagg HL.

---

### Official Review · Reviewer_xTCU · 2025-11-01

**Soundness:** 3
**Presentation:** 3
**Contribution:** 3
**Rating:** 6
**Confidence:** 3

**Summary:**

This paper presents Cronus, an LLM inference system for heterogeneous GPUs that partially disaggregates prefill to dynamically balance workloads. Cronus executes part of the prefill stage on low-end GPUs and the remaining prefill and decode stage on high-end GPUs, working asynchronously. Evaluation results show Cronus improves throughput and P99 TTFT and TBT over other parallelization techniques.

**Strengths:**

- Interesting disaggregation approach for heterogeneous GPUs, backed by careful analysis

**Weaknesses:**

- No ablation study is presented. It would be interesting to show how the design choices (especially the partial prefill length) would affect the overall performance. With the current evaluation, it is unclear whether the proposed design optimally balances the workloads between weak and strong GPU instances.
- No or little improvement in throughput compared to DP baseline (Table 2). Although it has a better P99 of TTFT/TBT, DP might be a better solution depending on the SLO requirements. For example, TTFT/TBT SLO requirements can be tens of seconds/hundreds of milliseconds [1, 2], and DP is better in those cases.

### References

- [1] https://arxiv.org/abs/2407.00079
- [2] https://arxiv.org/abs/2408.12757

**Questions:**

- Is it possible to quantify the labels in Table 1 in some way? Instead of qualitative labels like small/medium/large or high/low, it would be helpful to provide some approximation.
- Why do you model prefill times as a linear function of prefill context length (e.g., in Equation 2)? Does the computation not scale to the square of context length due to self-attention?
- Why do you use vLLM version 0.6.1.post2, which was released more than one year ago? Will there be any difference with the newer version of vLLM, especially the V1 architecture?

---

> ### Author Response · Authors · 2025-11-29
>
> **Q: Is it possible to quantify the labels in Table 1 in some way? Instead of qualitative labels like small/medium/large or high/low, it would be helpful to provide some approximation.**
>
> **A:** We have comparisons of different approaches’ TBT, TTFT, and throughput in our evaluation section. The results related to the load imbalance are in the appendix.
>
> **Q: Why do you model prefill times as a linear function of prefill context length (e.g., in Equation 2)? Does the computation not scale to the square of context length due to self-attention?**
>
> **A:** Based on our experiment, when the input length is relatively short (<10k), linear function is sufficient to model prefill batch latency. The quadratic term is only important when the context length is significantly large.
>
> **Q: Why do you use vLLM version 0.6.1.post2, which was released more than one year ago? Will there be any difference with the newer version of vLLM, especially the V1 architecture?**
>
> **A:** We developed and implemented our algorithm in October 2024. vLLM version 0.6.1 is what we had at the time.

---

### Official Review · Reviewer_fGMx · 2025-11-04

**Soundness:** 3
**Presentation:** 3
**Contribution:** 2
**Rating:** 4
**Confidence:** 4

**Summary:**

This paper tries to enhance the PD-disaggregation by executing partial of the prefill on the prefill instance and executing the remaining prefill and the decoding on the decoding instance. It aims to solve the load balance problem of the vanilla PD approach. It also studies how to estimate the execution time of prefill to make the workload more balanced between instances. Evaluation shows that this method can achieve better TTFT and TBT performance than different baselines.

**Strengths:**

1. It focuses on an important problem of PD-disaggregation.
2. It presents a different method than the vanilla PD method.

**Weaknesses:**

1. The motivation is weak as for using the split prefill to solve the imbalance problem of PD.
2. The method is relatively simple, and not practical enough for the industrial workloads and environment.
3. The analysis of different parallel methods is not solid. And the discussion of the related work is weak.
4. The balancer does not consider the length of the decoding.

**Questions:**

1. Given that PD is usually used for the industrial scenarios for better TTFT and TBT, why not using different numbers of GPUs for prefill and decoding directly to solve the imbalance problem? Note the industrial scenario will usually serve the models with much more than 2 GPUs. This method can be easier for the deployment and maintenance.
2. Given that the decoding length is not known ahead, even splitting the prefill on different instance, it still cannot achieve a perfect balance. How to address this problem?
3. In figure 2, why the remaining prefill is chunked, but the first partial prefill is not?
4. There lacks a discussion of the effort to address the imbalance problem in the related work.

---

> ### Author Response · Authors · 2025-11-29
>
> **Q: Given that PD is usually used for the industrial scenarios for better TTFT and TBT, why not using different numbers of GPUs for prefill and decoding directly to solve the imbalance problem? Note the industrial scenario will usually serve the models with much more than 2 GPUs. This method can be easier for the deployment and maintenance.**
>
> **A:** If a heterogeneous GPU cluster has a perfect ratio of two types of GPUs, we can assign one type of GPU (type A) to prefill and another type of GPU (type B) to decode then scale the number of prefill and decode instances to match the throughput. But in reality, it's possible we don’t have enough of one type of GPU to scale. As a result, the original PD will result in imbalance between prefill and decode.
>
> Cronus is designed to scale to X high-end GPUs and Y low-end GPUs. Because of hardware accessibility limitations, we could not conduct several experiments we intended in evaluation before submission. We have done experiments of more configurations and will add these results into the camera ready paper.
>
> In particular, we have run experiments on different numbers of high-end GPUs and low-end GPUs: with 2 A10+1 A100, and with 1 A10+2 A100. We used data parallelism in partial prefill instances and chunked prefill instances to allow more than one GPUs in these instances. To balance the throughput of partial prefill instances and chunked prefill instances, we simply adjusted the Balancer so that the ratio of partial prefill time over the chunked prefill time of a request matches the ratio of the number of chunked prefill instances and the number of partial prefill instances. In both configurations, Cronus showed the same advantage over other configurations mentioned in the paper.
>
> In more detail, in experiments with 2 A10+1 A100, Cronus reduces the TTFT P99 by up to 43% and TBT P99 by up to 40% compared to DP, and increases throughput by 750% compared to disagg LH and by 167% compared to disagg HL;  In experiments with 1 A10+2 A100, Cronus reduces the TTFT P99 by up to 33% and TBT P99 by up to 53% compared to DP, and increases throughput by 264% compared to disagg LH and by 41% compared to disagg HL.
>
> **Q: Given that the decoding length is not known ahead, even splitting the prefill on different instance, it still cannot achieve a perfect balance. How to address this problem?**
>
> **A:** In the design of the Cronus, we assume the decode is underloaded in the high-end GPUs, so the maximum throughput is limited by prefill. The balancer will guarantee the prefill is processed at maximum throughput.
> When decode instances are overloaded, prefill instances will continue to process prefill of requests until their KV cache buffer is full.
>
> **Q: In figure 2, why the remaining prefill is chunked, but the first partial prefill is not?**
>
> **A:** The remaining prefill is batched with decode in the chunked prefill instance. It is chunked to reduce P99 TBT. The first partial prefill is not chunked, because it is not batched with decode.

---

### Meta-Review · Area_Chair_a2Ro · 2025-12-31

**Summary:**

The paper introduces Cronus, a novel LLM inference system designed for heterogeneous GPU clusters that utilize partially disaggregated prefill to dynamically balance workloads. Reviewers generally recognized the importance of the problem (LLM serving on heterogeneous clusters). However, significant concerns remain regarding the completeness and breadth of the evaluation, particularly:

**Lack of Comparison to State-of-the-Art:** The baselines evaluated (DP, PP, simple disaggregation) were considered oversimplifications, and lack of comparison against sophisticated disaggregation methods like Splitwise and DistServe.

**Scalability and Generalization:** The evaluation was limited to setups with only two GPUs (e.g., A100+A10, A100+A30) Reviewers questioned the system's scalability to larger clusters, more than two GPU types, and the feasibility of its load-balancing approach in more complex topologies. The authors provided limited post-rebuttal data for 2~A10+1~A100 and 1~A10+2~A100.

**Methodological Weaknesses:** Concerns were raised about the simplicity and practicality of the method for industrial workloads, the lack of an ablation study to justify the partial prefill length, the lack of considering decoding length in the balancer, and the use of a linear model for prefill time estimation, which neglects the quadratic self-attention term for long contexts.

**Throughput vs. Baselines:** One reviewer noted the throughput improvement over DP was marginal in the original submission's evaluation.

**Reviewer Concerns:**

**Addressed Concerns:**
Why the remaining prefill is chunked but the partial prefill is not: Authors explained that the remaining prefill is batched with decode and chunked to reduce P99 TBT. The first partial prefill is not batched with decode, hence it is not chunked.

**Partially Addressed:**
System scalability to more than two GPUs/nodes: Authors added post-submission experimental results for 2~A10+1~A100 and 1~A10+2~A100, demonstrating the system's advantage in these configurations. They also stated Cronus is designed to scale to X high-end and Y low-end GPUs.

Linear modeling of prefill time despite quadratic self-attention: Authors stated that the linear model is sufficient for relatively short input lengths (10^k), where the quadratic term is not significantly large.

**Outstanding Concerns:**
Lack of evaluation against state-of-the-art disaggregation methods (e.g., Splitwise, DistServe)

The difficulty of balancing due to unknown decoding length: Authors clarified that the design assumes decode is underloaded on high-end GPUs, limiting throughput by prefill. The balancer focuses on maximizing prefill throughput, and prefill instances will stop if the KV cache buffer on the decode instance is full.

The robustness of the Balancer's predictive models to errors or inaccurate estimations

**Reviewer Scores:**

Reviewer fGMx (Current: 4): 4 to 6, as the rebuttal clarified prefill chunking and added initial scaling evidence, slightly easing the practicality/scaling concerns.

Reviewer XTCU (Current: 6): Remains 6, since the rebuttal justified the linear-time model and older vLLM choice but missing ablations and overall throughput concerns still stand.

Reviewer rtB8 (Current: 2): Remain 2, as the rebuttal addressed TP support conceptually, clarified the PPI batch-size limit, and provided initial scaling results, though SOTA comparison and evaluation scope remain weak.

Reviewer URKn (Current: 0): 0 to 2, because the “0” was due to a format violation and while technical merit may be around ~3 the rebuttal still leaves major concerns on Balancer overhead, robustness, and workload diversity.

---

### Decision · Program_Chairs · 2026-01-26

Reject